# NEIGHBOR2SEQ: DEEP LEARNING ON MASSIVE GRAPHS BY TRANSFORMING NEIGHBORS TO SEQUENCES

## ABSTRACT

Modern graph neural networks (GNNs) use a message passing scheme and have achieved great success in many fields. However, this recursive design inherently leads to excessive computation and memory requirements, making it not applicable to massive real-world graphs. In this work, we propose the Neighbor2Seq to transform the hierarchical neighborhood of each node into a sequence. This novel transformation enables the subsequent use of general deep learning operations, such as convolution and attention, that are designed for grid-like data. Therefore, our Neighbor2Seq naturally endows GNNs with the efficiency and advantages of deep learning operations on grid-like data by precomputing the Neighbor2Seq transformations. In addition, our Neighbor2Seq can alleviate the over-squashing issue suffered by GNNs based on message passing. We evaluate our method on a massive graph, with more than 111 million nodes and 1.6 billion edges, as well as several medium-scale graphs. Results show that our proposed method is scalable to massive graphs and achieves superior performance across massive and medium-scale graphs.

## 1 INTRODUCTION

Graph neural networks (GNNs) have shown effectiveness in many fields with rich relational structures, such as citation networks (Kipf & Welling, 2016; Veličković et al., 2018), social networks (Hamilton et al., 2017), drug discovery (Gilmer et al., 2017; Stokes et al., 2020), physical systems (Battaglia et al., 2016), and point clouds (Wang et al., 2019). Most current GNNs follow a message passing scheme (Gilmer et al., 2017; Battaglia et al., 2018), in which the representation of each node is recursively updated by aggregating the representations of its neighbors. Various GNNs (Li et al., 2016; Kipf & Welling, 2016; Veličković et al., 2018; Xu et al., 2019) mainly differ in the forms of aggregation functions.

Real-world applications usually generate massive graphs, such as social networks. However, message passing methods have difficulties in handling such large graphs as the recursive message passing mechanism leads to prohibitive computation and memory requirements. To date, sampling methods (Hamilton et al., 2017; Ying et al., 2018; Chen et al., 2018a;b; Huang et al., 2018; Zou et al., 2019; Zeng et al., 2020; Gao et al., 2018; Chiang et al., 2019; Zeng et al., 2020) and precomputing methods (Wu et al., 2019; Rossi et al., 2020; Bojchevski et al., 2020) have been proposed to scale GNNs on large graphs. While the sampling methods can speed up training, they might result in redundancy, still incur high computational complexity, lead to loss of performance, or introduce bias (see Section 2.2). Generally, precomputing methods can scale to larger graphs as compared to sampling methods as recursive message passing is still required in sampling methods.

In this work, we propose the Neighbor2Seq that transforms the hierarchical neighborhood of each node to a sequence in a precomputing step. After the Neighbor2Seq transformation, each node and its associated neighborhood tree are converted to an ordered sequence. Therefore, each node can be viewed as an independent sample and is no longer constrained by the topological structure. This novel transformation from graphs to grid-like data enables the use of mini-batch training for subsequent models. As a result, our models can be used on extremely large graphs, as long as the Neighbor2Seq step can be precomputed.

As a radical departure from existing precomputing methods, we consider the hierarchical neighborhood of each node as an ordered sequence. The order information corresponds to hops between nodes. As a result of our Neighbor2Seq transformation, generic deep learning operations for grid-like data, such as convolution and attention, can be applied in subsequent models. In addition, our Neighbor2Seq can alleviate the over-squashing issue (Alon & Yahav, 2020) suffered by current GNNs. Experimental results indicate that our proposed method can be used on a massive graph, where most current methods cannot be applied. Furthermore, our method achieves superior performance as compared with previous sampling and precomputing methods.

## 2 ANALYSIS OF CURRENT GRAPH NEURAL NETWORK METHODS

We start by defining necessary notations. A graph is formally defined as $\mathcal{G} = (V, E)$, where $V$ is the set of nodes and $E \subseteq V \times V$ is the set of edges. We use $n = |V|$ and $m = |E|$ to denote the numbers of nodes and edges, respectively. The nodes are indexed from 1 to $n$. We consider a node feature matrix $\boldsymbol{X} \in \mathbb{R}^{n \times d}$, where each row $\boldsymbol{x}_i \in \mathbb{R}^d$ is the $d$-dimensional feature vector associated with node $i$. The topology information of the graph is encoded in the adjacency matrix $\boldsymbol{A} \in \mathbb{R}^{n \times n}$, where $\boldsymbol{A}_{(i,j)} = 1$ if an edge exists between node $i$ and node $j$, and $\boldsymbol{A}_{(i,j)} = 0$ otherwise.

### 2.1 GRAPH NEURAL NETWORKS VIA MESSAGE PASSING

There are two primary deep learning methods on graphs (Bronstein et al.); those are, spectral methods and spatial methods. The spectral method in Bruna et al. (2014) extends convolutional neural networks (LeCun et al., 1989) to the graph domain based on the spectrum of the graph Laplacian. The main limitation of spectral methods is the high complexity. ChebNet (Defferrard et al., 2016) and GCN (Kipf & Welling, 2016) simplify the spectral methods and can be understood from the spatial perspective. In this work, we focus on the analysis of the current mainstream spatial methods. Generally, most existing spatial methods, such as ChebNet (Defferrard et al., 2016), GCN (Kipf & Welling, 2016), GG-NN (Li et al., 2016), GAT (Veličković et al., 2018), and GIN (Xu et al., 2019), can be understood from the message passing perspective (Gilmer et al., 2017; Battaglia et al., 2018). Specifically, we iteratively update node representations by aggregating representations from its immediate neighbors. These message passing methods have been shown to be effective in many fields. However, they have inherent difficulties when applied on large graphs due to their excessive computation and memory requirements, as described in Section 2.2.

### 2.2 GRAPH NEURAL NETWORKS ON LARGE GRAPHS

The above message passing methods are often trained in full batch. This requires the whole graph, *i.e.*, all the node representations and edge connections, to be in memory to allow recursive message passing on the whole graph. Usually, the number of neighbors grows very rapidly with the increase of receptive field. Hence, these methods cannot be applied directly on large-scale graphs due to the prohibitive requirements on computation and memory. To enable deep learning on large graphs, two families of methods have been proposed; those are methods based on sampling and precomputing.

To circumvent the recursive expansion of neighbors across layers, sampling methods apply GNNs on a sampled subset of nodes with mini-batch training. Sampling methods can be further divided into three categories. First, node-wise sampling methods perform message passing for each node in its sampled neighborhood. This strategy is first proposed in GraphSAGE (Hamilton et al., 2017), where neighbors are randomly sampled. This is extended by PinSAGE (Ying et al., 2018), which selects neighbors based on random walks. VR-GCN (Chen et al., 2018a) further proposes to use variance reduction techniques to obtain a convergence guarantee. Although these node-wise sampling methods can reduce computation, the remaining computation is still very expensive and some redundancy might have been introduced, as described in Huang et al. (2018). Second, layer-wise sampling methods sample a fixed number of nodes for each layer. In particular, FastGCN (Chen et al., 2018b) samples a fixed number of nodes for each layer independently based on the degree of each node. AS-GCN (Huang et al., 2018) and LADIES (Zou et al., 2019) introduce between-layer dependencies during sampling, thus alleviating the loss of information. Layer-wise sampling methods can avoid the redundancy introduced by node-wise sampling methods. However, the expensive sampling algorithms that aim to ensure performance may themselves incur high computational cost,

as pointed out in Zeng et al. (2020). Third, graph-wise sampling methods build mini-batches on sampled subgraphs. Specifically, LGCN (Gao et al., 2018) proposes to leverage mini-batch training on subgraphs selected by Breadth-First-Search algorithms. ClusterGCN (Chiang et al., 2019) conducts mini-batch training on sampled subgraphs that are obtained by a graph clustering algorithm. GraphSAINT (Zeng et al., 2020) proposes to derive subgraphs by importance-sampling and introduces normalization techniques to eliminate biases. These graph-wise sampling methods usually have high efficiency. The main limitation is that the nodes in a sampled subgraph are usually clustered together. This implies that two distant nodes in the original graph usually cannot be fed into the GNNs in the same mini-batch during training, potentially leading to bias in the trained models.

The second family of methods for enabling GNNs training on large graphs are based on procomputing. Specifically, SGC (Wu et al., 2019) removes the non-linearity between GCN layers, resulting in a simplification as $\boldsymbol{Y} = \text{softmax}(\hat{\boldsymbol{A}}^L \boldsymbol{X} \boldsymbol{W})$. In this formulation, $\hat{\boldsymbol{A}} = \tilde{\boldsymbol{D}}^{-\frac{1}{2}} \tilde{\boldsymbol{A}} \tilde{\boldsymbol{D}}^{-\frac{1}{2}}$ is the symmetrically normalized adjacency matrix, $\tilde{\boldsymbol{A}} = \boldsymbol{A} + \boldsymbol{I}$ is the adjacency matrix with self-loops, $\tilde{\boldsymbol{D}}$ is the corresponding diagonal node degree matrix with $\tilde{\boldsymbol{D}}_{(i,i)} = \sum_j \tilde{\boldsymbol{A}}_{(i,j)}$, $L$ is the size of receptive field (*i.e.*, the number of considered neighboring hops), which is the same as a $L$-layer GCN, $\boldsymbol{Y}$ is the output of the softmax classifier. Since there is no learnable parameters in $\hat{\boldsymbol{A}}^L \boldsymbol{X}$, this term can be precomputed as a feature pre-processing step. Similarly, SIGN (Rossi et al., 2020) applies an inception-like model to the precomputed features $\hat{\boldsymbol{A}}^\ell \boldsymbol{X}$ for $\ell \in \{1, \cdots, L\}$, where $L$ is the predefined size of receptive field. Instead of precomputing the smoothing features as in SGC and SIGN, PPRGo (Bojchevski et al., 2020) extends the idea of PPNP (Klicpera et al., 2018) by approximately precomputing the personalized PageRank (Page et al., 1999) matrix, thereby enabling model training on large graphs using mini-batches. Generally, the precomputing methods can scale to larger graphs than sampling methods because the latter still needs to perform the recursive message passing during training. Differing from these precomputing methods, we consider the hierarchical neighborhood of each node as an ordered sequence, thus retaining the useful information about hops between nodes and enabling subsequent powerful and efficient operations.

## 3 THE PROPOSED NEIGHBOR2SEQ METHOD AND ANALYSIS

In this section, we describe our proposed method, known as Neighbor2Seq, which transforms the hierarchical neighborhood of each node into an ordered sequence, thus enabling the subsequent use of general deep learning operations. We analyze the scalability of our method (See Section 3.5) and describe how our method can alleviate the over-squashing issue suffered by current message passing methods (See Section 3.6).

### 3.1 OVERVIEW

As described in Section 2.1, existing message passing methods recursively update each node's representation by aggregating information from its immediate neighbors. Hence, what these methods aim at capturing for each node is essentially its corresponding hierarchical neighborhood, *i.e.*, the neighborhood tree rooted at current node, as illustrated in Figure 1 (b). In this work, we attempt to go beyond the message passing scheme to overcome the limitations mentioned in Section 2. We propose to capture the information of this hierarchical neighborhood by transforming it into an ordered sequence, instead of recursively squashing it into a fixed-length vector. Our proposed method is composed of three steps. First, we transform a neighborhood to a sequence for each node. Second, we apply a normalization technique to the derived sequence features. Third, we use general deep learning operations, *i.e.*, convolution and attention, to learn from these sequence features and then make predictions for nodes. In the following, we describe these three steps in detail.

### 3.2 NEIGHBOR2SEQ: TRANSFORMING NEIGHBORHOODS TO SEQUENCES

The basic idea of Neighbor2Seq is to transform the hierarchical neighborhood of each node to an ordered sequence by integrating the features of nodes in each layer of the neighborhood tree. Following the notations defined in Section 2, we let $\boldsymbol{z}_0^i, \boldsymbol{z}_1^i, \cdots, \boldsymbol{z}_L^i$ denote the resulting sequence for node $i$, where $L$ is the height (*i.e.*, the number of hops we consider) of the neighborhood tree rooted at node $i$. $\boldsymbol{z}_\ell^i \in \mathbb{R}^d$ denotes the $\ell$-th feature of the sequence. The length of the resulting sequence for

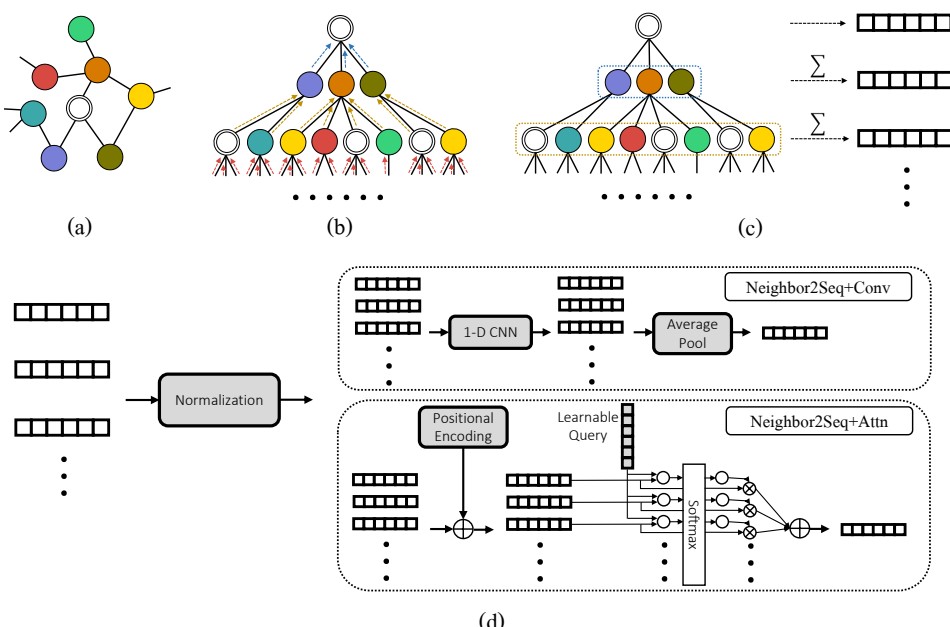

(a)  (b)  (c)

(d)

Figure 1: (a) An illustration of the original graph. The current node is denoted as two concentric circles. (b) Message passing in the neighborhood tree. (c) Our proposed Neighbor2Seq. (d) Our proposed models: Neighbor2Seq+Conv and Neighbor2Seq+Attn.

each node is $L + 1$. Formally, for each node $i$, our Neighbor2Seq can be expressed as

$$\boldsymbol{z}_\ell^i = \sum_{j=1}^n w(i, j, \ell) \boldsymbol{x}_j, \quad \forall \ell \in \{0, 1, 2, \cdots, L\}, \tag{1}$$

where $w(i, j, \ell)$ denotes the number of walks with length $\ell$ between node $i$ and $j$. $n$ is the number of nodes in the graph. We define $w(i, j, 0)$ as 1 for $j = i$ and 0 otherwise. Hence, $\boldsymbol{z}_0^i$ is the original node feature $\boldsymbol{x}_i$. Intuitively, $\boldsymbol{z}_\ell^i$ is obtained by computing a weighted sum of features of all nodes with walks of length $\ell$ to $i$, and the numbers of qualified walks are used as the corresponding weights. Our Neighbor2Seq is illustrated in Figure 1 (c). Note that the derived sequence has meaningful order information, indicating the hops between nodes. After we obtain ordered sequences from the original hierarchical neighborhoods, we can use generic deep learning operations to learn from these sequences, as detailed below.

### 3.3 NORMALIZATION

Since the number of nodes in the hierarchical neighborhood grows exponentially as the hop number increases, different layers in the neighborhood tree have drastically different numbers of nodes. Hence, feature vectors of a sequence computed by Equation (1) have very different scales. In order to make the subsequent learning easier, we propose a layer to normalize the sequence features. We use a normalization technique similar to layer normalization (Ba et al., 2016). In particular, each feature of a sequence is normalized based on the mean and the standard deviation of its own feature values. Formally, our normalization process for each node $i$ can be written as

$$\boldsymbol{y}_\ell^i = \boldsymbol{W}_{\boldsymbol{\ell}} \boldsymbol{z}_\ell^i, \quad \boldsymbol{o}_\ell^i = \frac{\boldsymbol{y}_\ell^i - \mu_\ell^i}{\sigma_\ell^i} \odot \boldsymbol{\gamma}_\ell + \boldsymbol{\beta}_\ell, \quad \forall \ell \in \{0, 1, 2, \cdots, L\}$$

$$\mu_\ell^i = \frac{1}{d'} \sum_{c=1}^{d'} \boldsymbol{y}_\ell^i[c], \quad \sigma_\ell^i = \sqrt{\frac{1}{d'} \sum_{c=1}^{d'} (\boldsymbol{y}_\ell^i[c] - \mu_\ell^i)^2}. \tag{2}$$

We first apply a linear transformation $\boldsymbol{W}_{\boldsymbol{\ell}} \in \mathbb{R}^{d' \times d}$ to produce a low-dimensional representation $\boldsymbol{y}_\ell^i \in \mathbb{R}^{d'}$ for the $\ell$-th feature of the sequence, since the original feature dimension $d$ is usually

large. $\mu_\ell^i \in \mathbb{R}$ and $\sigma_\ell^i \in \mathbb{R}$ are the mean and standard deviation of the corresponding representation $\boldsymbol{y}_\ell^i$. $\boldsymbol{\gamma}_\ell \in \mathbb{R}^{d'}$ and $\boldsymbol{\beta}_\ell \in \mathbb{R}^{d'}$ denote the learnable affine transformation parameters. $\odot$ denotes the element-wise multiplication. Note that the learnable parameters in this normalization layer is associated with $\ell$, implying that each feature of the sequence is normalized separately. Using this normalization layer, we obtain the normalized feature vector $\boldsymbol{o}_\ell^i \in \mathbb{R}^{d'}$ for every $\ell \in \{0, 1, 2, \cdots, L\}$.

### 3.4 NEIGHBOR2SEQ+CONV AND NEIGHBOR2SEQ+ATTN

After obtaining an ordered sequence for each node, we can view each node in the graph as a sequence of feature vectors. We can use general deep learning techniques to learn from these sequences. In this work, we propose two models, namely Neighbor2Seq+Conv and Neighbor2Seq+Attn, in which convolution and attention are applied on the sequences of each node.

As illustrated in Figure 1 (d), Neighbor2Seq+Conv applies a 1-D convolutional neural network to the sequence features and then use an average pooling to yield a representation for the sequence. Formally, for each node $i$,

$$\left(\hat{\boldsymbol{o}}_0^i, \hat{\boldsymbol{o}}_1^i, \cdots, \hat{\boldsymbol{o}}_L^i\right) = \text{CNN}\left(\boldsymbol{o}_0^i, \boldsymbol{o}_1^i, \cdots, \boldsymbol{o}_L^i\right), \quad \boldsymbol{r}^i = \frac{1}{L+1} \sum_{\ell=0}^{L} \hat{\boldsymbol{o}}_\ell^i, \tag{3}$$

where $\text{CNN}(\cdot)$ denotes a 1-D convolutional neural network. $\boldsymbol{r}^i$ denotes the obtained representation of node $i$ that is used as the input to a linear classifier to make predictions for this node. Specifically, we implement $\text{CNN}(\cdot)$ as a 2-layer convolutional neural network composed of two 1-D convolutions. The kernel size is set according to the length of input sequence. The activation function between layers is ReLU (Krizhevsky et al., 2012).

Incorporating attention is another natural idea to learn from sequences. As shown in Figure 1 (d), Neighbor2Seq+Attn uses an attention mechanism (Bahdanau et al., 2015) to integrate sequential feature vectors in order to derive a representation. Unlike convolutional neural networks, the vanilla attention mechanism cannot make use of the order of the sequence. Hence, we add positional encodings (Vaswani et al., 2017) to the features such that the position information of different features in the sequence can be incorporated. Formally, for each node $i$, we add positional encoding for each feature in the sequence as

$$\boldsymbol{k}_\ell^i = \boldsymbol{o}_\ell^i + \boldsymbol{p}_\ell^i, \quad \boldsymbol{p}_\ell^i[m] = \begin{cases} \sin\left(\dfrac{\ell}{10000^{\frac{2n}{d'}}}\right) & m = 2n \\[4mm] \cos\left(\dfrac{\ell}{10000^{\frac{2n}{d'}}}\right) & m = 2n+1 \end{cases}. \tag{4}$$

The positional encoding for $\ell$-th feature of node $i$ is denoted as $\boldsymbol{p}_\ell^i \in \mathbb{R}^{d'}$. $m \in \{1, 2, \cdots, d'\}$ is the dimensional index. Intuitively, a position-dependent vector is added to each feature such that the order information can be captured. Then we use the attention mechanism with learnable query (Yang et al., 2016) to combine these sequential feature vectors to obtain the final representations $\boldsymbol{r}^i$ for each node $i$. Formally,

$$\boldsymbol{r}^i = \sum_{\ell=0}^{L} \alpha_\ell^i \boldsymbol{k}_\ell^i, \quad \alpha_\ell^i = \frac{\exp(\boldsymbol{k}_\ell^{i^T} \boldsymbol{q})}{\sum_{\ell=0}^{L} \exp(\boldsymbol{k}_\ell^{i^T} \boldsymbol{q})}. \tag{5}$$

$\boldsymbol{q} \in \mathbb{R}^{d'}$ is the learnable query vector that is trained along with other model parameters. The derived representation $\boldsymbol{r}^i$ will be taken as the input to a linear classifier to make prediction for node $i$.

### 3.5 ANALYSIS OF SCALABILITY

**Precomputing Neighbor2Seq.** A well-known fact is that the value of $w(i, j, \ell)$ in Equation (1) can be obtained by computing the power of the original adjacency matrix $\boldsymbol{A}$. Following GCN, we add self-loops to make each node connected to itself. Concretely, $w(i, j, \ell) = \tilde{\boldsymbol{A}}_{(i,j)}^\ell$. Hence, the Neighbor2Seq can be implemented by computing the matrix multiplications $\tilde{\boldsymbol{A}}^\ell \boldsymbol{X}$ for $\forall \ell \in \{0, 1, 2, \cdots, L\}$. Since there is no learnable parameters in the Neighbor2Seq step, these matrix multiplications can be precomputed sequentially for large graphs on CPU platforms with large memory.

This can be easily precomputed because the matrix $\tilde{A}$ is usually sparse. For extremely large graphs, this precomputation can even be performed on distributed systems.

**Enabling mini-batch training.** After we obtain the precomputed sequence features, each node in the graph corresponds to a sequence of feature vectors. Therefore, each node can be viewed as an independent sample. That is, we are no longer restricted by the original graph connectivity anymore. Then, we can randomly sample from all the training nodes to conduct mini-batch training. This is more flexible and unbiased than sampling methods as reviewed in Section 2.2. Our mini-batches can be randomly extracted over all nodes, opening the possibility that any pair of nodes can be sampled in the same mini-batch. In contrast, mini-batches in sampling methods are usually restricted by the fixed sampling strategies. This advantage opens the door for subsequent model training on extremely large graphs, as long as the corresponding Neighbor2Seq step can be precomputed.

**Computational complexity comparison.** We compare our methods with several existing sampling and precomputing methods in terms of computational complexity. We let $L$ denote the number of hops we consider. For simplicity, we assume the feature dimension $d$ is fixed for all layers. For sampling methods, $s$ is the number of sampled neighbors for each node. The computation of Neighbor2Seq+Conv mainly lies in the linear transformation (*i.e.*, $\mathcal{O}(Ld^2n)$) in the normalization step and the 1-D convolutional neural networks (*i.e.*,

Table 1: Comparison of computational complexity for precomputing and forward pass corresponding to an entire epoch.

| Method | Precomputing | Forward Pass |
|---|---|---|
| GCN | - | $\mathcal{O}(Ldm + Ld^2n)$ |
| GraphSAGE | $\mathcal{O}(s^L n)$ | $\mathcal{O}(s^L d^2 n)$ |
| ClusterGCN | $\mathcal{O}(m)$ | $\mathcal{O}(Ldm + Ld^2n)$ |
| GraphSAINT | $\mathcal{O}(sn)$ | $\mathcal{O}(Ldm + Ld^2n)$ |
| SGC | $\mathcal{O}(Ldm)$ | $\mathcal{O}(d^2n)$ |
| SIGN | $\mathcal{O}(Ldm)$ | $\mathcal{O}(Ld^2n)$ |
| Neighbor2Seq+Conv | $\mathcal{O}(Ldm)$ | $\mathcal{O}((Ld^2 + Lkd^2)n)$ |
| Neighbor2Seq+Attn | $\mathcal{O}(Ldm)$ | $\mathcal{O}((Ld^2 + Ld)n)$ |

$\mathcal{O}(Lkd^2n)$, where $k$ is the kernel size). Hence, the computational complexity for the forward pass of Neighbor2Seq+Conv is $\mathcal{O}((Ld^2 + Lkd^2)n)$. Neighbor2Seq+Attn has a computational complexity of $\mathcal{O}((Ld^2 + Ld)n)$ because the attention mechanism is more efficient than 1-D convolutional neural networks. As shown in Table 1, the forward pass complexities of precomputing methods, including our Neighbor2Seq+Conv and Neighbor2Seq+Attn, are all linear with respect to the number of nodes $n$ and do not depend on the number of edges $m$. Hence, the training processes of our models are computationally efficient.

### 3.6 ALLEVIATING THE OVER-SQUASHING ISSUE

An inherent problem in message passing methods is known as the over-squashing (Alon & Yahav, 2020). In particular, recursively propagating information between neighbors creates a bottleneck because the number of nodes in the receptive field grows exponentially with the number of layers. This bottleneck causes the over-squashing issue; that is, information from the exponentially-growing receptive field is compressed into fixed-length vectors. Consequently, message passing methods fail to capture the message flowing from distant nodes and performs poorly when long-range information is essential for the prediction tasks. Note that the over-squashing issue is not identical to the over-smoothing issue. Over-smoothing is related to the phenomenon that node representations converge to indistinguishable limits when the number of layers increases (Li et al., 2018; Wu et al., 2019; NT & Maehara, 2019; Liu et al., 2020a; Oono & Suzuki, 2020; Cai & Wang, 2020; Chen et al., 2020). The virtual edges added in Gilmer et al. (2017) and recent non-local aggregations (Pei et al., 2020; Liu et al., 2020b) can be viewed as attempts to alleviate the over-squashing issue by incorporating distant nodes. Another study (Ma et al., 2019) considers message passing along all possible paths between two nodes, instead of propagating information between neighbors.

Our Neighbor2Seq can alleviate the over-squashing issue because we transform the exponentially-growing nodes in hierarchical neighborhoods into an ordered sequence, instead of recursively squashing them into a fixed-size vector. With our Neighbor2Seq, capturing long-range information on graphs becomes similar to achieving this on sequence data, such as texts.

## 4 DISCUSSIONS

### 4.1 INFORMATION LOSS IN NEIGHBOR2SEQ

As shown in Figure 1 (c), Neighbor2Seq obtains the sequence by integrating features of nodes in each layer of the neighborhood tree. This transformation may lose the cross-layer dependency

information in the tree. Specifically, the Neighbor2Seq ignores the identities of nodes that each walk passes through and only considers what are the nodes in each layer of the neighborhood tree. Nevertheless, this information can neither be captured by message passing methods because the aggregation is usually permutation-invariant. This implies that messages from different neighbors cannot be distinguished, as pointed in Pei et al. (2020). According to our experimental results in Table 5, our models without this information can outperform message passing methods, such as GCN. It is intriguing to have an in-depth exploration of whether such information is useful and how it can be captured.

## 4.2 RELATIONS WITH THE WEISFEILER-LEHMAN HIERARCHY

As shown in Xu et al. (2019), most of current GNNs are at most as powerful as the Weisfeiler-Lehman (WL) graph isomorphism test (Weisfeiler & Lehman, 1968) in distinguishing graph structures. Our Neighbor2Seq is still under the WL hierarchy because the neighborhood tree used to obtain the sequence is indeed the one that the WL test uses to distinguish different graphs. We would be interested in exploring how Neighbor2Seq can be extended to go beyond the WL hierarchy as a future direction.

## 4.3 BRIDGING THE GAP BETWEEN GRAPH AND GRID-LIKE DATA

The main difference between graph and grid-like data lies in the notion and properties of locality. Specifically, the numbers of neighbors differ for different nodes, and there is no order information among the neighbors of a node in graphs. These are the main obstacles preventing the use of generic deep learning operations on graphs. Our Neighbor2Seq is an attempt to bridge the gap between graph and grid-like data. Base on our Neighbor2Seq, many effective strategies for grid-like data can be naturally transferred to graph data. These include self-supervised learning and pre-training on graphs (Hu et al., 2019; Velickovic et al., 2019; Sun et al., 2019; Hassani & Khasahmadi, 2020; You et al., 2020; Hu et al., 2020b; Qiu et al., 2020; Jin et al., 2020).

We notice an existing work AWE Ivanov & Burnaev (2018) which also embed the information in graph as a sequence. In order to avoid confusion, we make a clarification about the fundamental and significant differences between AWE and our Neighbor2Seq. First, AWE produces a sequence embedding for the entire graph, while our Neighbor2Seq yields a sequence embedding for each node in the graph. Second, each element in the obtained sequence in AWE is the probability of an anonymous walk embedding. In our Neighbor2Seq, each feature vector in the obtained sequence for one node is computed by summing up the features of all nodes in the corresponding layer of the neighborhood tree. This point distinguishes these two methods fundamentally.

## 5 EXPERIMENTAL STUDIES

### 5.1 EXPERIMENTAL SETUP

**Datasets.** We evaluate our proposed models on 1 massive-scale graph and 4 medium-scale graphs using node classification tasks. The massive-scale graph *ogbn-papers100M* provided by the Open Graph Benchmark (OGB) (Hu et al., 2020a) is the existing largest benchmark dataset for node classification. Medium-scale graphs include *ogbn-products* (Hu et al., 2020a), *Reddit* (Hamilton et al., 2017), *Yelp* Zeng et al. (2020), and *Flickr* Zeng et al. (2020). These tasks cover inductive and transductive settings. The statistics of these datasets are summarized in Table 2. The detailed description of these datasets are provided in Appendix A.1.

Table 2: Statistics of datasets. "m" denotes multi-label classification.

| Dataset | Scale | #Nodes | #Edges | Avg. Deg. | #Features | #Classes | Train/Val/Test |
|---|---|---|---|---|---|---|---|
| *ogbn-papers100M* | Massive | $111,059,956$ | $1,615,685,872$ | 29 | 128 | 172 | 0.78/0.08/0.14 |
| *ogbn-products* | Medium | $2,449,029$ | $61,859,140$ | 51 | 100 | 47 | 0.08/0.02/0.90 |
| *Reddit* | Medium | $232,965$ | $11,606,919$ | 50 | 602 | 41 | 0.66/0.10/0.24 |
| *Yelp* | Medium | $716,857$ | $6,997,410$ | 10 | 300 | 100(m) | 0.75/0.10/0.15 |
| *Flickr* | Medium | $89,250$ | $899,756$ | 10 | 500 | 7 | 0.50/0.25/0.25 |

**Implementation.** We implemented our methods using Pytorch (Paszke et al., 2017) and Pytorch Geometric (Fey & Lenssen, 2019). For our proposed methods, we conduct the precomputation on

the CPU, after which we train our models on a GeForce RTX 2080 Ti GPU. We perform a grid search for the following hyperparameters: *number of hops L*, *batch size*, *learning rate*, *hidden dimension $d'$*, *dropout rate*, *weight decay*, and *convolutional kernel size k*. The chosen hyperparameters for our Neighbor2Seq+Conv and Neighbor2Seq+Attn are summarized in Appendix A.2 for reproducibility.

## 5.2 RESULTS ON MASSIVE-SCALE GRAPHS

Since *ogbn-papers100M* is a massive graph with more than 111 million nodes and 1.6 billion edges, most existing methods have difficulty handling such a graph. We consider three baselines that have available results evaluated by OGB: Multilayer Perceptron (MLP), Node2Vec (Grover & Leskovec, 2016), and SGC (Wu et al., 2019). The results under transductive setting is reported in Table 3. Following OGB, we report accuracies for all models on training, validation, and test sets. The previous state-of-the-art result on *ogbn-papers100M* is ob-

Table 3: Results on *ogbn-papers100M* in terms of classification accuracy (in percent). The reported accuracy is averaged over 10 random runs. Note that existing sampling methods cannot scale to this massive graph. During precomputation, both SGC and our models have to randomly remove 40% edges to avoid a memory overflow on CPU. This implies that the performance could be further improved if more advanced precomuting platform is used.

| Method | Training | Validation | Test |
|---|---|---|---|
| MLP | $54.84_{\pm 0.43}$ | $49.60_{\pm 0.29}$ | $47.24_{\pm 0.31}$ |
| Node2vec | - | $58.07_{\pm 0.28}$ | $55.60_{\pm 0.23}$ |
| SGC | $67.54_{\pm 0.43}$ | $66.48_{\pm 0.20}$ | $63.29_{\pm 0.19}$ |
| Neighbor2Seq+Conv | $\mathbf{69.87}_{\pm 0.81}$ | $\mathbf{67.46}_{\pm 0.16}$ | $\mathbf{64.04}_{\pm 0.22}$ |
| Neighbor2Seq+Attn | $\underline{68.83}_{\pm 0.30}$ | $\underline{66.90}_{\pm 0.10}$ | $\underline{63.59}_{\pm 0.17}$ |

tained by the precomputing method SGC. Our models outperform the baselines consistently in terms of training, validation, and test, which demonstrates the expressive power and the generalization ability of our method on massive graphs.

## 5.3 RESULTS ON MEDIUM-SCALE GRAPHS

We also evaluate our models on medium-scale graphs, thus enabling comparison with more existing works. We conduct transductive learning on *ogbn-products*, a medium-scale graph from OGB. We also conduct inductive learning on *Reddit*, *Yelp*, and *Flickr*, which are frequently used for inductive learning by the community. The following baselines are considered: MLP, Node2Vec (Grover & Leskovec, 2016), GCN (Kipf & Welling, 2016), SGC Wu et al. (2019), GraphSAGE (Hamilton et al., 2017), FastGCN (Chen et al., 2018b), VR-GCN (Chen et al., 2018a), AS-GCN (Huang et al., 2018), ClusterGCN (Chiang et al., 2019), GraphSAINT (Zeng et al., 2020), and SIGN (Rossi et al., 2020).

Table 4: Results on *ogbn-products* in terms of classification accuracy (in percent). The reported accuracy is averaged over 10 random runs. Obtaining the results of GCN requires a GPU with 33GB of memory.

| Method | Training | Validation | Test |
|---|---|---|---|
| MLP | $84.03_{\pm 0.93}$ | $75.54_{\pm 0.14}$ | $61.06_{\pm 0.08}$ |
| Node2vec | $93.39_{\pm 0.10}$ | $90.32_{\pm 0.06}$ | $72.49_{\pm 0.10}$ |
| GCN | $93.56_{\pm 0.09}$ | $92.00_{\pm 0.03}$ | $75.64_{\pm 0.21}$ |
| GraphSAGE | $92.96_{\pm 0.07}$ | $91.70_{\pm 0.09}$ | $78.70_{\pm 0.36}$ |
| ClusterGCN | $93.75_{\pm 0.13}$ | $92.12_{\pm 0.09}$ | $78.97_{\pm 0.33}$ |
| GraphSAINT | $92.71_{\pm 0.14}$ | $91.62_{\pm 0.08}$ | $79.08_{\pm 0.24}$ |
| SGC | $92.60_{\pm 0.10}$ | $91.19_{\pm 0.06}$ | $72.46_{\pm 0.27}$ |
| SIGN | $\mathbf{96.92}_{\pm 0.46}$ | $\mathbf{93.10}_{\pm 0.08}$ | $77.60_{\pm 0.13}$ |
| Neighbor2Seq+Conv | $\underline{95.32}_{\pm 0.10}$ | $\underline{92.92}_{\pm 0.05}$ | $\mathbf{79.67}_{\pm 0.16}$ |
| Neighbor2Seq+Attn | $92.82_{\pm 0.14}$ | $92.20_{\pm 0.02}$ | $\underline{79.35}_{\pm 0.17}$ |

Table 5: Results for inductive learning on three datasets in terms of F1-micro score. The reported score is averaged over 10 random runs. The results of baselines are partially obtained from Zeng et al. (2020); Rossi et al. (2020).

| Method | *Reddit* | *Flickr* | *Yelp* |
|---|---|---|---|
| GCN | $0.933_{\pm 0.000}$ | $0.492_{\pm 0.003}$ | $0.378_{\pm 0.001}$ |
| FastGCN | $0.924_{\pm 0.001}$ | $0.504_{\pm 0.001}$ | $0.265_{\pm 0.053}$ |
| VR-GCN | $0.964_{\pm 0.001}$ | $0.482_{\pm 0.003}$ | $0.640_{\pm 0.002}$ |
| AS-GCN | $0.958_{\pm 0.001}$ | $0.504_{\pm 0.002}$ | - |
| GraphSAGE | $0.953_{\pm 0.001}$ | $0.501_{\pm 0.013}$ | $0.634_{\pm 0.006}$ |
| ClusterGCN | $0.954_{\pm 0.001}$ | $0.481_{\pm 0.005}$ | $0.609_{\pm 0.005}$ |
| GraphSAINT | $0.966_{\pm 0.001}$ | $0.511_{\pm 0.001}$ | $\mathbf{0.653}_{\pm 0.003}$ |
| SGC | $0.949_{\pm 0.000}$ | $0.502_{\pm 0.001}$ | $0.358_{\pm 0.006}$ |
| SIGN | $\mathbf{0.968}_{\pm 0.000}$ | $0.514_{\pm 0.001}$ | $0.631_{\pm 0.003}$ |
| Neighbor2Seq+Conv | $\underline{0.967}_{\pm 0.000}$ | $\mathbf{0.527}_{\pm 0.003}$ | $\underline{0.647}_{\pm 0.003}$ |
| Neighbor2Seq+Attn | $\underline{0.967}_{\pm 0.000}$ | $\underline{0.523}_{\pm 0.002}$ | $\underline{0.647}_{\pm 0.001}$ |

The *ogbn-products* dataset is challenging because the splitting is not random. The splitting procedure is more realistic. The nodes (*i.e.*, products) are sorted according to their sales ranking and the top 8% nodes are used for training, next 2% for validation, and the rest 90% for testing. This matches the real-world application where manual labeling is prioritized to important nodes and models are subsequently used to make prediction on less important nodes. Hence, *ogbn-products* is an ideal benchmark dataset to improve out-of-distribution prediction. As shown in Table 4, our Neighbor2Seq+Conv and Neighbor2Seq+Attn outperfom baselines on test set (*i.e.*, 90% nodes), which further demonstrates the generalization ability of our method.

The results on inductive tasks are summarized in Table 5. On *Reddit*, our models perform better than all sampling methods and achieve the competitive result as SIGN. On *Flickr*, our models obtain significantly better results. Specifically, our Neighbor2Seq+Conv outperforms the previous state-of-the-art models by an obvious margin. Although our models perform not as good as GraphSAINT on *Yelp*, we outperform other sampling methods and the precomputing model SIGN consistently on this dataset.

## 5.4 COMPARISONS OF COMPUTATIONAL EFFICIENCY

In order to show the computational efficiency, we conduct an empirical comparison with existing methods in terms of time consuming during preprocessing, training, and inference. We consider the following representative sampling methods and precomputing methods: ClusterGCN Chiang et al. (2019), GraphSAINT Zeng et al. (2020), SGC Wu et al. (2019), and SIGN Rossi et al. (2020). The comparison is performed on *ogbn-products* and the similar trend can be observed on other datasets. As demonstrated in Table 6, our approaches, like existing precomputing methods, are more computationally efficient than sampling methods in terms of training and inference. Compared with existing precomputing methods, our methods achieve a better balance between performance and efficiency.

Table 6: Computational efficiency in terms of preprocessing, training (per epoch), and inference times (in seconds) on *ogbn-products*. The reported time is averaged over 10 runs.

| Method | Preprocessing ($\downarrow$) | Training ($\downarrow$) | Inferenece ($\downarrow$) | Test Accuracy ($\uparrow$) |
|---|---|---|---|---|
| ClusterGCN | $44.15_{\pm 0.77}$ | $11.87_{\pm 0.84}$ | $87.03_{\pm 0.24}$ | $78.97_{\pm 0.33}$ |
| GraphSAINT | $80.78_{\pm 3.5}$ | $4.29_{\pm 0.48}$ | $107.26_{\pm 0.94}$ | $79.08_{\pm 0.24}$ |
| SGC | $153.36_{\pm 3.6}$ | $0.15_{\pm 0.01}$ | $1.08_{\pm 0.01}$ | $72.46_{\pm 0.27}$ |
| SIGN | $151.47_{\pm 3.5}$ | $1.22_{\pm 0.02}$ | $2.93_{\pm 0.06}$ | $77.60_{\pm 0.13}$ |
| Neighbor2Seq+Conv | $153.42_{\pm 3.2}$ | $4.09_{\pm 0.12}$ | $31.52_{\pm 1.44}$ | $79.67_{\pm 0.16}$ |
| Neighbor2Seq+Attn | $153.42_{\pm 3.2}$ | $2.67_{\pm 0.08}$ | $31.24_{\pm 0.61}$ | $79.35_{\pm 0.17}$ |

## 5.5 ABLATION STUDY ON ORDER INFORMATION

Table 7: Comparison of models with and without capturing order information. Neighbor2Seq+Attn w/o PE denotes the Neighbor2Seq+Attn without adding positional encoding.

| Model | Order | *ogbn-papers100M* | *ogbn-products* | *Reddit* | *Flickr* | *Yelp* |
|---|---|---|---|---|---|---|
| Neighbor2Seq+Conv | ✓ | $\mathbf{64.04}_{\pm 0.22}$ | $\mathbf{79.67}_{\pm 0.16}$ | $\mathbf{0.967}_{\pm 0.000}$ | $\mathbf{0.527}_{\pm 0.003}$ | $\mathbf{0.647}_{\pm 0.003}$ |
| Neighbor2Seq+Attn | ✓ | $63.59_{\pm 0.17}$ | $79.35_{\pm 0.17}$ | $\mathbf{0.967}_{\pm 0.000}$ | $0.523_{\pm 0.002}$ | $\mathbf{0.647}_{\pm 0.001}$ |
| Neighbor2Seq+Attn w/o PE | ✗ | $63.61_{\pm 0.09}$ | $78.54_{\pm 0.25}$ | $0.965_{\pm 0.000}$ | $0.521_{\pm 0.003}$ | $0.646_{\pm 0.001}$ |

Intuitively, the order information in the sequence obtained by Neighbor2Seq indicates the hops between nodes. Hence, we conduct an ablation study to verify the significance of this order information. We remove the positional encoding in Neighbor2Seq+Attn, leading to a model without the ability to capture the order information. The comparison is demonstrated in Table 7. Note that Neighbor2Seq+Attn and Neighbor2Seq+Attn w/o PE have the same number of parameters. Hence, Comparing the results of these two models, we can conclude that the order information is usually necessary. Neighbor2Seq+Conv and Neighbor2Seq+Attn both can capture the order information. There are two possible reasons why Neighbor2Seq+Conv performs better. First, Neighbor2Seq+Conv has more learnable parameters than Neighbor2Se+Attn, which only has a learnable query. Second, the convolutional neural network in Neighbor2Seq+Conv can additionally investigate the dependencies between feature dimensions because each feature dimension of the output depends on every feature dimension of the input.

## 6 CONCLUSIONS AND OUTLOOK

In this work, we propose Neighbor2Seq, for transforming the heirarchical neighborhoods to ordered sequences. Neighbor2Seq enables the subsequent use of powerful general deep learning operations, leading to the proposed Neighbor2Seq+Conv and Neighbor2Seq+Attn. Our models can be deployed on massive graphs and trained efficiently. The extensive expriments demonstrate the scalability and the promising performance of our method. As discussed in Section 4, based on our Neighbor2Seq, several significant directions can be further explored in the future research.

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

# A APPENDIX

## A.1 DATASET DESCRIPTIONS

*ogbn-papers100M* (Hu et al., 2020a) is the existing largest benckmark dataset for node classification. It is a directed citation graph of 111 million papers indexed by Microsoft Academic Graph (MAG) (Wang et al., 2020). For simplicity, it is converted to an undirected graph in baselines and our method. Each node is a paper and each directed edge indicates that one paper cites another one. Each node is associated with a 128-dimensional feature vector obtained by averaging the word2vec (Mikolov et al., 2013) embeddings of words in its title and abstract. Among the node set, approximately 1.5 millione of them are ARXIV papers, each of which has a label with one of ARXIV's subject areas. The rest nodes (*i.e.*, non-ARXIV papers) are not associated with label information. The task is to leverage the entire citation graph to infer the labels of the ARXIV papers. The time-based splitting is used as the splitting strategy. To be more specifical, the training nodes are all ARXIV papers published until 2017, while the validation nodes are the ARXIV papers published in 2018, and the ARXIV papers published since 2019 are treated as test nodes.

*ogbn-products* (Hu et al., 2020a) is an undirected Amazon product co-purchasing graph (Bhatia et al., 2016). Nodes denote products and edges between two nodes indicate that the corresponding products are purchased together. Node features are derived by extracting bag-of-words representations from the product descriptions. Further, a Principal Component Analysis is applied to these features to reduce the dimension to 100. The task is to predict the category of a product. A realistic splitting scheme is used in this data. Specifically, the products are firstly sorted according to their sales ranking, and then the top 10% products are used for training, next 2% for validation, and the rest for testing. This strategy matches the real-world situation where manual labeling is prioritized to important nodes and models are subsequently deployed to predict the less important ones.

*Reddit* (Hamilton et al., 2017), *Yelp* (Zeng et al., 2020), and *Flickr* (Zeng et al., 2020) are widely used datasets for inductive learning. During training, only the node features of training nodes and the edges between training nodes are visible. *Reddit* is a social netowork extracted from Reddit forum. Nodes represent posts and edges between two posts indicate the same user comments on both posts. Node features are fromed by GloVe CommonCrawl word vectors Pennington et al. (2014) of the posts. The task is to predict which community different posts belong to. The splitting is also time-based. *Yelp* is a social netowork constructed from Yelp website. Nodes are users and edges between two nodes indicate they are friends. Node features of users are obtained by the word2vec embeddings of their corresponding reviews. The task is to predict the categories of businesses reviewed by different users, which is multi-label classification task. *Flickr* is a social network based on Flickr, a photo sharing website. Nodes represent images and there is an edge between two nodes if two images share some common properties. The node features are fromed by the bag-of-words representations of the images. The task is to predict the category each image belongs to.

## A.2 HYPERPARAMETER CONFIGURATIONS

We conduct a grid search for hyperparameters. Table 8 summarizes the chosen hyperparameters for our models.

Table 8: The chosen hyperparameters for our models on all datasets.

| Model | Hyperparameter | *ogbn-papers100M* | *ogbn-products* | *Reddit* | *Flickr* | *Yelp* |
|---|---|---|---|---|---|---|
| Neighbor2Seq+Conv | *number of hops $L$* | 3 | 7 | 3 | 10 | 2 |
| | *hidden dimension $d'$* | 512 | 512 | 256 | 256 | 512 |
| | *convolutional kernel size $k$* | 5 | 7 | 5 | 7 | 3 |
| | *learning rate* | 5e-4 | 2e-5 | 8e-5 | 8e-4 | 5e-4 |
| | *batch size* | 12288 | 64 | 32768 | 24576 | 8192 |
| | *weight decay* | 5e-5 | 5e-5 | 0 | 5e-5 | 0 |
| | *dropout rate* | 0.25 | 0.5 | 0.5 | 0.5 | 0 |
| Neighbor2Seq+Attn | *number of hops $L$* | 3 | 7 | 3 | 10 | 2 |
| | *hidden dimension $d'$* | 512 | 512 | 256 | 256 | 512 |
| | *learning rate* | 5e-4 | 1e-3 | 2e-3 | 2e-3 | 5e-4 |
| | *batch size* | 12288 | 3072 | 32768 | 256 | 8192 |
| | *weight decay* | 5e-6 | 5e-5 | 0 | 5e-5 | 0 |
| | *dropout rate* | 0.25 | 0.5 | 0.5 | 0.5 | 0 |

