# OpenReview forum: "Neighbor2Seq: Deep Learning on Massive Graphs by Transforming Neighbors to Sequences"
_ICLR.cc/2021/Conference — Reject_

### Official Review · AnonReviewer1 · 2020-10-27
**Lacks discussions with prior works**

**Rating:** 5
**Confidence:** 4

**Review:**

Overall, the paper propose an interesting approach for computing node embeddings in a scalable way.
However, the contribution is incremental as the idea of embedding nodes using sequences is not new; moreover, the discussion with prior works is very weak.

Concretely, important related works are missing.
There is an ICML 2018 paper, "Anonymous Walk Embeddings" (https://arxiv.org/pdf/1805.11921.pdf). There are many following up papers as well. The idea of this line of works is to embedding node/graphs by random walk sequences. While the approaches are not exactly the same as  Neighbor2Seq, the idea of embedding nodes using sequences is not new. However, none of these works are mentioned in this paper. The authors should compare these papers in the experiments as the baselines. Without such comparison, it is hard to evaluate the performance gain of the proposed Neighbor2Seq.

Additionally, the discussion with prior works is very weak.
The paper misleads the readers by arguing "However, they have inherent difficulties when applied on large graphs due to their excessive computation and memory requirements" in Section 2.1. The paper refers
However, existing approaches can scale to huge graphs. For example in the PinSage paper (https://arxiv.org/pdf/1806.01973.pdf), they can scale their GNN model to 3 billion nodes. Failing to mentioning this point is misleading for the readers, and is exaggerating the motivation of this paper.

More comments:
1 The exact configuration of the Neighbor2Seq model is not mentioned. Providing an algorithm will greatly help.

2 I can imagine multiple ways for sampling sequences from a node's neighborhood. However, they are not discussed in the paper.

3 How does the expressive power of Neighbor2Seq compares with WL test? I assume Neighbor2Seq is theoretically less expressive than WL test.

4 I guess there will be failure cases of this kind of Neighbor2Seq, when the topology of node neighborhood structure matters for the final performance. For example, Neighbor2Seq probably won't perform well for graph isomorphism tests. I suggest mentioning potential limitations in the paper.

---

> ### Author Response · Authors · 2020-11-13
> **Our response to Reviewer #1 (Part 2)**
>
> Q3: The exact configuration of the Neighbor2Seq model is not mentioned. Providing an algorithm will greatly help.
>
> A3: We included the configurations in the appendix and mentioned it in the main text.
>
> Q4: I can imagine multiple ways for sampling sequences from a node's neighborhood. However, they are not discussed in the paper.
>
> A4: Since the hierarchical neighborhood is what GNN and WL test aim to capture, we transform the hierarchical neighborhood of each node into a sequence without applying any sampling to avoid unnecessary information loss.
>
> Q5: How does the expressive power of Neighbor2Seq compared with WL test? I assume Neighbor2Seq is theoretically less expressive than WL test.
>
> A5: Great point! As we discussed in Section 4.2, our Neighbor2Seq is still under the WL hierarchy since the neighborhood tree used to obtained the sequence is indeed the one that WL test uses to distinguish different graphs. Hence, Neighbor2Seq is as powerful as WL test if we can fully capture/distinguish the neighborhood tree during transforming. Furthermore, this is equivalent to can we fully capture/distinguish each layer of neighborhood tree. Our answer is yes (approximately). If we formulate the nodes in each layer of the neighborhood tree as a multiset, then our operation on this multiset is a summation, followed by a neural network. This design can approximate an injective function on a multiset. The proof could be similar to the proof of GIN[1].
>
> In summary, our Neighbor2Seq is under the WL hierarchy and could be as powerful as WL test (approximately). We hope this can address you concerns to some degree, and we will make the above clarification more professional in our final version. Thank you again for proposing this insightful question.
>
> Q6: I guess there will be failure cases of this kind of Neighbor2Seq, when the topology of node neighborhood structure matters for the final performance. For example, Neighbor2Seq probably won't perform well for graph isomorphism tests. I suggest mentioning potential limitations in the paper.
>
> A6: Please refer to the above A5 for our clarification about the relationship between Neighbor2Seq and WL test. We have mentioned limitations of Neighbor2Seq in the discussion (Section 4), including the information loss of Neighbor2Seq and the relations with WL hierarchy.
>
>
> Thank you again for these constructive and insightful questions. We hope we addressed your concerns. Hope we can have further discussion with you about these great points.
>
>
> [1] Xu et al.. How powerful are graph neural networks?. ICLR 2019.

---

> > ### Comment · AnonReviewer1 · 2020-11-24
> > **Reply to authors**
> >
> > I thank the authors for providing these detailed explanations.
> > It's good to see the authors included the discussion on AWE. They significantly expanded the discussions with other sampling based scalable GNN methods as well.
> > However, there still lacks an overall theoretical understandings on Neighbor2Seq. In addition, the authors acknowledge that Neighbor2Seq comes with limitations, however, no experiment has been done to illustrate that point.
> >
> > Nevertheless, I'm happy to see that the authors have reflected my suggestions; therefore, I'm happy to increase my score from 4 to 5.

---

> > > ### Author Response · Authors · 2020-11-24
> > > **Authors' response (2nd round)**
> > >
> > > Thank you for responding to us and raising the score.
> > >
> > > (1) We have to acknowledge that it would be challenging to provide strict theoretical support for Neighbor2Seq, currently. As described in our paper, we developed Neighbor2Seq for capturing the neighborhood tree, which is also the goal of existing popular message passing methods. As we know, there is no strict theoretic background for these message passing methods. However, they are shown to be effective empirically. Our Neighbor2Seq is also demonstrated to be effective and efficient.
> > >
> > > (2) As we discussed in Section 4.1, there is information loss during transforming. That is, Neighbor2Seq transformation may lose the cross-layer dependency information in the neighborhood tree. Nevertheless, this information can neither be captured by message passing methods because the aggregation in these method is usually permutation-invariant. This implies that messages from different neighbors cannot be distinguished, as pointed in [1]. We believe this universal limitation should be investigated more by the community in the future.
> > >
> > > Overall, based on general message passing methods, our Neighbor2Seq aims at scaling to massive-scale graphs with keeping effectiveness. According to our comprehensive experiments, our Neighbor2Seq achieves this significant goal.
> > >
> > > Thank you again for your helpful response. Hope we addressed your concerns.
> > >
> > > [1] Pei et al. Geom-GCN: Geometric Graph Convolutional Networks. ICLR 2020.

---

> ### Author Response · Authors · 2020-11-13
> **Our response to Reviewer #1 (Part 1)**
>
> Thank you for your constructive comments. We have revised our paper based on the reviewers’ insightful suggestions. We answer the questions below.
>
> Q1: Concretely, important related works are missing. There is an ICML 2018 paper, "Anonymous Walk Embeddings" (https://arxiv.org/pdf/1805.11921.pdf). There are many following up papers as well. The idea of this line of works is to embedding node/graphs by random walk sequences. While the approaches are not exactly the same as Neighbor2Seq, the idea of embedding nodes using sequences is not new. However, none of these works are mentioned in this paper. The authors should compare these papers in the experiments as the baselines. Without such comparison, it is hard to evaluate the performance gain of the proposed Neighbor2Seq.
>
> A1: Thank you for leading us to this great work. Based on the limitations of popular message passing methods, such as GCN, GAT, and GIN, we developed Neighbor2Seq to capture the hierarchical neighborhood that GNN and WL test try to capture. Hence, our attention is drawn by the literature in the community of graph neural networks, and we didn’t notice the work you mentioned.
>
> We have read this work in detail. Both this AWE and our Neighbor2Seq aim to use sequence(s) to encode the information contained in graph. we would like to make a clarification about the fundamental and significant differences between our Neighbor2Seq and AWE. (1) AWE produces a sequence embedding for the entire graph, while our Neighbor2Seq yield a sequence embedding for each node in the graph. (2) In AWE, each element in the obtained sequence is the probability of an anonymous walk embedding. In our Neighbor2Seq, each “token” in the obtained sequence for one node is computed by summing up the features of all nodes in the corresponding layer of the neighborhood tree. This point distinguishes these two methods fundamentally. (3) Due to (1), AWE is proposed for scalable graph-level tasks and evaluated on graph classification problems. Instead, our work aims to conduct node-level predictions on large-scale graphs. Hence, it is not natural to compare them empirically.
>
> We have added the above clarification to Section 4.3 in our revised version. We really appreciate you making us to be aware of these wonderful works!
>
> Q2: Additionally, the discussion with prior works is very weak. The paper misleads the readers by arguing "However, they have inherent difficulties when applied on large graphs due to their excessive computation and memory requirements" in Section 2.1. The paper refers However, existing approaches can scale to huge graphs. For example in the PinSage paper (https://arxiv.org/pdf/1806.01973.pdf), they can scale their GNN model to 3 billion nodes. Failing to mentioning this point is misleading for the readers, and is exaggerating the motivation of this paper.
>
> A2:
> Sorry for such unclarity. We would like to make this clearer.
>
> In our original version, in Section 2.1, “They” in the sentence "However, they have inherent difficulties when applied on large graphs due to their excessive computation and memory requirements" refer to the message passing methods reviewed in Section 2.1, such as GCN, GAT, GIN, etc. Then we described why this happens for these methods in the first several sentences in Section 2.2. That’s why we say “As described in Section 2.2” in the last sentence of Section 2.1.
>
> Next, in Section 2.2, we discussed the prior works that can scale to large graphs in a unified and comprehensive way. Starting with “To enable deep learning on large graphs, two families of methods have been proposed; those are methods based on sampling and precomputing. ”, we reviewed these existing methods on large graphs comprehensively, including GraphSAGE, PinSAGE, VR-GCN, FastGCN, As-GCN, LADIES, LGCN, ClusterGCN, GraphSAINT, SGC, SIGN, and PPRGo.
>
> Hope this can make it clearer and can addressed your concerns.

---

### Official Review · AnonReviewer3 · 2020-10-28
**need more clarification on the connection of the method and the issues it's trying to mitigate**

**Rating:** 5
**Confidence:** 4

**Review:**

The paper proposes a method called neighbor2seq that converts the hierarchical structure of the center node to a sequence during message passing in graph neural networks. The proposed method aims to mitigate the issue of excessive computation and memory requirement of training graph neural networks. The proposed models Neighbor2Seq+Conv and Neighbor2Seq+Attn are tested on several datasets including a large scale benchmark dataset (ogbn-papers100M). The result shows some improvement especially on ogbn-papers100M while the improvement is not very obvious on other datasets.

Strength:
1. The idea of converting the graph topology to a sequence that could be tackled with methods designed for grid-like data is good.
2. The proposed method shows improvement on a large scale dataset. There are also ablation studies that are useful to understand different components in the method, for example, removing the sequence information (position encoding here) clearly hurts the model performance. But it's surprising to see that Neighbor2Seq+Conv is better than Neighbor2Seq+Attn. I think there could be more information/experiment/insights added to explain why this is the case.

Weakness:
1. Since the proposed model is trying to mitigate the excessive computation and memory requirement issues, make it more efficient, and alleviate over-squashing, I'm expecting to see more in-depth analysis and empirical/theoretical justification with regard to those issues that the method is trying to solve. However, there's only some time complexity comparison and some argument about over-squashing. Can you provide the comparison of space complexity since memory requirement mitigation is one of your motivation? Also, Neighbor2Seq does not seem to improve upon the computation as compare with most of the methods listed in table 1? For the argument of alleviating over-squashing, can you provide more details? Why is converting to a sequence alleviating over-squashing? It seems that some sequence models also suffer from over-squashing, isn't it (trying to squash the whole sequence information into a single vector)? Please provide more explanation/analytical results, etc.
2. The experiment results do not seem to improve a lot compared with existing models for some datasets and tasks. It seems that Neighbor2Seq+Conv consistently performs better than Neighbor2Seq+Attn in all experiment. Can you provide analysis of the number of parameters of your proposed model (2 variants) compared with existing models? Since people usually see that attention based approach performs better, is it because conv based one has more parameters. Does the proposed method have more parameters than existing models in general? Can you also provide some empirical comparison on the training time of your model compared with baseline models?

---

> ### Author Response · Authors · 2020-11-12
> **Our response to Reviewer #3**
>
> We want to thank the reviewer for the insightful comments. We have revised our paper based on the reviewers’ helpful suggestions. We answer the questions as follows.
>
> Q1: Since the proposed model is trying to mitigate the excessive computation and memory requirement issues, make it more efficient, and alleviate over-squashing, I'm expecting to see more in-depth analysis and empirical/theoretical justification with regard to those issues that the method is trying to solve. However, there's only some time complexity comparison and some argument about over-squashing. Can you provide the comparison of space complexity since memory requirement mitigation is one of your motivation? Also, Neighbor2Seq does not seem to improve upon the computation as compare with most of the methods listed in table 1? For the argument of alleviating over-squashing, can you provide more details? Why is converting to a sequence alleviating over-squashing? It seems that some sequence models also suffer from over-squashing, isn't it (trying to squash the whole sequence information into a single vector)? Please provide more explanation/analytical results, etc.
>
> A1: Thanks for these great suggestions.
>
> (1) Yes, one of our motivations is to alleviate the memory overflow on large graphs by enabling mini-batch training using our Neighbor2Seq. Hence, memory cost is highly related to the batch size. For example, on ogbn-papers100M dataset, the largest batch size such that our Neighbor2Seq+Conv can be trained on a single GPU (11GB) is about $2^{17}=131072$.
>
> (2) We have added a Section 5.4 for the computational comparison with representative sampling and precomputing methods in our revised version. Our approaches are more computationally efficient than sampling methods in terms of training and inference. Compared with existing precomputing methods, our methods achieve a better balance between performance and efficiency. Please refer to Section 5.4 for details.
>
> (3) Our Neighbor2Seq can alleviate the over-squashing issue [1] because we make the information flow of capturing long-range dependencies easier. Specifically, we transform the exponentially growing nodes in the hierarchical neighborhood into an ordered sequence, instead of recursively squashing them into a fixed-size vector. To understand why general message passing methods that squash exponentially increasing features into a fixed-size vector suffer from over-squashing issue, let’s suppose node *v* needs the information from a distant node *u* that is $k$-hop away from *v*.  Then neighbor-wise message passing methods need at least $k$ layers to capture such information, which will lead to over-squashing issue. Our Neighbor2Seq can alleviate this issue because it provides a “shortcut” for capturing the information of *u*. Instead of squashing exponentially increasing features into a fixed-size vector, Neighbor2Seq create a sequence of vector for each node to capture information from neighbors at different hops.
> Yes, some sequence models also suffer from over-squashing (or bottleneck) issue. Our Neighbor2Seq can **alleviate** this issue to some degree. How to overcome this issue is a significant future direction.
>
> Q2: The experiment results do not seem to improve a lot compared with existing models for some datasets and tasks. It seems that Neighbor2Seq+Conv consistently performs better than Neighbor2Seq+Attn in all experiment. Can you provide analysis of the number of parameters of your proposed model (2 variants) compared with existing models? Since people usually see that attention based approach performs better, is it because conv based one has more parameters. Does the proposed method have more parameters than existing models in general?
>
> A2:
>
> (1) Our models achieve better or competitive results on all datasets compared with strong baselines. Currently, there does not exist a method that can perform overwhelmingly on all these datasets. Therefore, we believe the experimental results can strongly demonstrate the effectiveness of our methods.
>
> (2) As we analyzed in Section 5.5, there are two possible reasons why Neighbor2Seq+Conv usually performs better. First, Neighbor2Seq+Conv has more learnable parameters than Neighbor2Se+Attn, which only has a learnable query. Second, the convolutional neural network in Neighbor2Seq+Conv can additionally investigate the dependencies between feature dimensions because each feature dimension of the output depends on every feature dimension of the input.
>
> (3) The numbers of parameters of the proposed methods are comparable with previous methods, such as GCN. The additional parameters only come from the convolutional kernels and learnable attention query, which are relatively few.
>
> Thank you again for the comprehensive and helpful comments. Hope your concerns are addressed.
>
> [1] Alon et al.. On the bottleneck of graph neural networks and its practical implications. Preprint 2020.

---

> > ### Comment · AnonReviewer3 · 2020-11-24
> > **thank you for the response**
> >
> > I appreciate your response and your adding section 5.4. However, I see that computation efficiency (table 6) does not overwhelmingly support your statement in the motivation. It seems to me that SIGN is much more efficient in terms of training and inference while only slightly worse in test accuracy. Moreover, SIGN is actually better in some of the test datasets. Similar conclusion can be drawn for GraphSAINT. I agree that there might be a balance between complexity and performance.

---

> > > ### Author Response · Authors · 2020-11-24
> > > **Authors' response (2nd round)**
> > >
> > > We really thank the reviewer for responding to us.
> > >
> > > (1) We would like to point out that there does not exist a method that can perform overwhelmingly on all these datasets, Currently.
> > >
> > > (2) As described in the analysis of existing methods for large-scale graphs in Section 2.2, our models, like other precomputing methods (such as SGC and SIGN), have more efficient training and inference process. Compared to the SOTA precomputing method SIGN, our model achieves a better balance between performance and efficiency. To make it more convincing, we compare the performance between SIGN and Neighbor2Seq+Conv directly as follows. Our method outperforms SIGN consistently on $3$ among $4$ datasets by obvious margins. We believe the improvements are significant, with considering the afforadable additional computation.
> > >
> > > |Method------------------|ogbn-products|Reddit        |Flickr   ------              |Yelp                     |
> > >
> > > |SIGN----------------------|-----77.60-----|---**96.8** ---|-----51.4    ----  |63.1                    |
> > >
> > > |Neighbor2Seq+Conv|**79.67(+2.07)**|96.7(-0.10)|**52.7(+1.3)**|**64.7(+1.6)**|
> > >
> > >
> > > Thank you again for your response and helpful comments. Hope your concerns are addressed.

---

### Official Review · AnonReviewer4 · 2020-10-29
**Limited novelty and contribution**

**Rating:** 4
**Confidence:** 3

**Review:**

The authors proposed a neighborhood to sequence construction & pre-training approach to handling graph representation learning on large graphs.

The merits of this work include:

1. provide a decent attempt toward solving the computation bottleneck for representation learning on large graphs;
2. discusses and shows the benefits of the pre-training based on (unsupervised) sequence learning.

The limitations, on the other hand, is obvious:

1. Pre-training is a relatively standard technique for representation learning on large graphs, and the schema proposed in this paper has very limited novelty;
2. The application of attention mechanism for sequence learning is also a standard practice that has been widely adopted in this domain;
3. The only part where this paper may distinguish itself from the previous work is how the sequence is constructed from the neighborhood. However, minimal theoretical discussion and empirical ablation study are provided to reveal the guarantees & benefits of the proposed method.

Therefore, the limited novelty and contribution of this paper clearly outweigh its merits.

=======================================

After reviewing the response from the authors, I decide to change my evaluation score and confidence score.

---

> ### Author Response · Authors · 2020-11-11
> **Our response to Reviewer #4**
>
> Thank you for the feedback.
>
> **We believe you might have some misunderstanding of our paper. Our proposed approach has nothing to do with pretraining.** Our Neighbor2Seq transforms the hierarchical neighborhood of each node into an ordered sequence by precomputing the transformation from a graph to sequences (on CPU), and then performs CNN or attention mechanism to learn from these sequences in minibatch training. Hence, our method can be scaled to extremely large graphs since we don’t need to load the whole graph into GPU memory during training.
>
> **Our Neighbor2Seq might facilitate future research about pretraining and self-supervised learning on graphs.** As discussed in Section 4.3 in our paper, our Neighbor2Seq can be viewed as an attempt to bridge the gap between graph data and grid-like data. Based on our Neighbor2Seq, many effective strategies for grid-like data can be naturally transferred to graph data, especially the effective pretraining and self-supervised learning methods developed on sequence data by the NLP community.
>
> Thank you again for your comments. Hope this clarification can solve your concerns.

---

> > ### Comment · AnonReviewer4 · 2020-11-24
> > **Thanks for the response**
> >
> > I want to thank the authors for providing response and clarifying my misunderstandings. However, I am still not convinced that there is enough novelty in this paper. I have raised the evaluation score and decreased my confidence score.

---

> > > ### Author Response · Authors · 2020-11-24
> > > **Authors' response (2nd round)**
> > >
> > > Thank you for your response.
> > >
> > > Our Neighbor2Seq novelly transforms a neighborhood tree into a sequence by precomputing, thus enabling us to train the subsequent general deep learning operations on massive-scale graphs in a mini-batch fashion. The comprehensive experiments also show the effectiveness and efficiency of our approach.
> > >
> > > It would be helpful for us if you can elaborate on your point regarding the novelty.
> > >
> > > Thank you again for your time and response. Hope we have addressed your concerns.

---

### Official Review · AnonReviewer2 · 2020-10-30
**review for "Neighbor2Seq"**

**Rating:** 7
**Confidence:** 4

**Review:**


# Summary

This paper proposed a simple graph neural network architecture that is easy to scale up and perform stochastic training. Instead of performing message passing as commonly used GNN, this paper first performs weighted combinations of node features per each hop of the neighbors of a center node, and then performs either CNN or attention mechanism to aggregate the features and obtain center node embedding. Since the feature aggregation can be performed offline, and the computation can easily be decomposed and stochastic training is straightforward, the method can easily scale up to graphs with 10M nodes. Experiments on median size or large size graphs show the comparable or better performance than alternatives.

# Pros

- The idea is simple and easy to implement, while being effective at the same time.
- The new design of the architecture
- Experiments on large scale graphs are convincing.

# Cons

- some comparisons are incomplete
- Not sure how general this approach would be
- time/memory cost can be reported and compared


# Details

Overall I lean towards accepting the paper.

This paper provides a simple yet efficient and effective approach for graph node embedding calculation. It enjoys similar computation efficiency as the SGC, but is a bit more expressive in the design, where the CNN or attention is introduced on top of the per-hop embeddings. I like such a simple design that adds the expressiveness without too much additional cost.

It is also good to see the large scale experiments on OGB graphs.

There are several aspects that can potentially be improved:

1. The SGC results are not presented in Table 4 or Table 5. It is necessary to include, as this might be the most relevant/comparable baseline.

2.  Would this approach be useful for graph classification? I understand that the main purpose of this approach is scalability, but it would also be good to know the potential limitation on its parameterization. Also it would be more comprehensive to see the results on small benchmarks like Cora, Pubmed, etc,. This is mainly to polish the paper and get a better understanding for users. It would be fine if the results are worse on these small graphs.

3. The runtime (during preprocessing, stochastic training, etc) and memory cost can be reported.

# Questions

I’d like to see the replies to my questions above.

# Improvement

It would be good to include additional experiments as mentioned above, to make the paper more comprehensive.

---

> ### Author Response · Authors · 2020-11-12
> **Our response to Reviewer #2**
>
> Thank you for your constructive comments. We have revised our paper based on the reviewers’ suggestions. We answer the questions below.
>
>
> Q1: The SGC results are not presented in Table 4 or Table 5. It is necessary to include, as this might be the most relevant/comparable baseline.
>
> A1: Thank you for pointing out this. Yes, SGC is a significant baseline and should be compared comprehensively. We have added the results of SGC on more datasets. The performance of SGC and our methods is compared below (The detailed results can be found in our revised version). Overall, our approaches consistently outperform SGC on all datasets by significant margins.
>
> |Method                       |ogbn-papers100M|ogbn-products|Reddit|Flickr|Yelp|
>
> |SGC	                          |63.29                       |72.46                 |0.949 |0.502|0.358|
>
> |Neighbor2Seq+Conv|64.04                       |79.67                  |0.967 |0.527|0.647|
>
> |Neighbor2Seq+Attn |63.59                       |79.35                  |0.967 |0.523|0.647|
>
>
> Q2: Would this approach be useful for graph classification? I understand that the main purpose of this approach is scalability, but it would also be good to know the potential limitation on its parameterization. Also it would be more comprehensive to see the results on small benchmarks like Cora, Pubmed, etc,. This is mainly to polish the paper and get a better understanding for users. It would be fine if the results are worse on these small graphs.
>
> A2:
>
> (1) Great point! In graph classification tasks, each graph can be considered as a sample and the size of each graph is usually relatively small, such as molecular graphs. Hence, for graph classification tasks, we can naturally apply stochastic training. However, in node classification tasks, nodes cannot be viewed as independent samples since they are connected in particular way. That’s why we only develop our method on node-level classification tasks. For graph classification tasks, most existing methods can be trained directly using mini-batches. Hence, for graph classification tasks, the community focus on developing models with powerful expressivity, such as GIN[1].
>
> (2) We have conducted experiments on three citation datasets. The results of our methods are similar to the popular methods like GCN. To be specific, our Neighbor2Seq+Conv achieve an accuracy of 81.7 on Cora. Since these datasets have some quality issues (as described in [2]) and cannot show the scalability of our methods, we don’t include these results in our paper.
>
>
> Q3: The runtime (during preprocessing, stochastic training, etc) and memory cost can be reported.
>
> A3:
>
> (1) Thank you for this great suggestion. We have added a Section 5.4 for the computational comparison with representative sampling and precomputing methods. Our approaches, like existing precomputing methods, are more computationally efficient than sampling methods in terms of training and inference. Compared with existing precomputing methods, our methods achieve a better balance between performance and efficiency. Please refer to Section 5.4 for details.
>
> (2) In terms of memory cost, this is highly related to the batch size. For example, on ogbn-papers100M dataset, the largest batch size such that our Neighbor2Seq+Conv can be trained on a single GPU (11GB) is about $2^{17}=131072$.
>
>
> Thank you again for your helpful comments. Hope we addressed your concerns.
>
> [1] Xu et al.. How powerful are graph neural networks?. ICLR 2019.
>
> [2] Hu et al.. Open graph benchmark: datasets for machine learning on graphs. NeurIPS 2020.

---

> > ### Comment · AnonReviewer2 · 2020-11-24
> > **RE: Our response to Reviewer #2**
> >
> > Thanks a lot for the reply and additional experiments and numbers.
> > I'll stay positive and keep my current evaluation.

---

> > > ### Author Response · Authors · 2020-11-24
> > > **Authors' response (2nd round)**
> > >
> > > Thank you for the helpful comments and positive evaluation.

---

### Author Response · Authors · 2020-11-19
**Response to all reviewers**

Dear reviewers,

Thanks for the constructive comments. We revised our paper according to your helpful suggestions and provided comprehensive responses to the proposed concerns. Please kindly check it. We hope we have addressed the concerns raised by the reviewers. Thank you.

---

### Decision · Program_Chairs · 2021-01-07
**Final Decision**

**Decision:**

Reject

**Comment:**

This paper presents a way to aggregate and precompute node features on a graph to enable fast parallel training of neural models on massive graphs for various node prediction tasks.  We have seen quite a few papers in this line of work (precompute node features without training, and then treat the nodes as independent during training) recently and this is a continuation in this trend.

Most reviewers lean toward rejection.  The main concern is the lack of novelty and the marginally better results reported in the experiments.  In some sense, the proposed method could be thought of as replacing the concatenation operation of node features across multiple hops used in the SIGN paper with a sequence model, either conv + pool, or attention.  Given this, the novelty of this paper is indeed a bit limited.  Additionally, it is unclear why this sequence model perspective is better than concatenation, which should be in principle more expressive and in practice faster and more efficient (as also reported in Table 6).

I recommend rejecting this paper, but do encourage the authors to position their work better with respect to prior work and really consider what’s the defining advantage of their approach compared to alternatives, like SIGN.